# Capturing implicit hierarchical structure in 3D biomedical images with self-supervised hyperbolic representations

**Joy Hsu**∗
Department of Computer Science
Stanford University
joycj@stanford.edu

**Jeffrey Gu**∗
ICME
Stanford University
jeffgu@stanford.edu

**Gong Her Wu**
Department of Bioengineering
Stanford University
wukon@stanford.edu

**Wah Chiu**
Department of Bioengineering
Stanford University
wahc@stanford.edu

**Serena Yeung**
Department of Biomedical Data Science
Stanford University
syyeung@stanford.edu

## Abstract

We consider the task of representation learning for unsupervised segmentation of 3D voxel-grid biomedical images. We show that models that capture *implicit* hierarchical relationships between subvolumes are better suited for this task. To that end, we consider encoder-decoder architectures with a hyperbolic latent space, to explicitly capture hierarchical relationships present in subvolumes of the data. We propose utilizing a 3D hyperbolic variational autoencoder with a novel gyroplane convolutional layer to map from the embedding space back to 3D images. To capture these relationships, we introduce an essential self-supervised loss—in addition to the standard VAE loss—which infers approximate hierarchies and encourages implicitly related subvolumes to be mapped closer in the embedding space. We present experiments on both synthetic data and biomedical data to validate our hypothesis.

## 1 Introduction

Advances in biomedical imaging techniques such as cryogenic electron tomography (cryo-ET) and magnetic resonance imaging (MRI) have resulted in an ever-increasing amount of 3D biomedical image data. In these data domains, a growing body of work shows that, when provided with labels, machine learning models achieve good performance on many tasks [Çiçek et al., 2016, Milletari et al., 2017, Dou et al., 2017, Falk et al., 2019]. However, these labels, especially for segmentation, are very costly as they often have to be provided by experts in the appropriate field. Consequently, supervised learning and even semi-supervised learning remain limited in this setting as (1) tasks and domains are very diverse, making it intractable for experts to provide labelled data for all problems; and (2) experts can only label objects they already know, restricting the potential for scientific discovery using machine learning methods. In this work, we tackle the task of unsupervised segmentation in 3D biomedical image data.

Our key insight is that 3D biomedical images have inherent hierarchical structure. For example, in the cryo-ET domain, an image of a cell has a hierarchy that at the highest level comprises the entire cell; at a finer level comprises organelles such as the mitochondria and nucleus; and at an even

---

∗These authors contributed equally.

35th Conference on Neural Information Processing Systems (NeurIPS 2021).

finer level comprises sub-structures such as the nucleolus of a nucleus or protein machineries within organelles. Such types of hierarchies are present in many types of biomedical images (e.g., nested anatomical structures within MRI images). We hypothesize that in the unsupervised setting, models that are able to encode this internal hierarchical structure will provide better data representations for downstream tasks. To that end, we propose learning representations based on embedding subvolumes of 3D images in hyperbolic space.

In contrast to traditional Euclidean embeddings, hyperbolic embeddings better preserve hierarchical relationships present in the data. Hyperbolic representations have been proposed as a continuous way to represent hierarchical data, due to their ability to embed trees with arbitrarily low error [Sarkar, 2011]. A recent line of work utilizes hyperbolic representations to model hierarchical data across domains ranging from natural language word taxonomies [Nickel and Kiela, 2017, 2018] and graphs [Nickel and Kiela, 2017, Mathieu et al., 2019, Ovinnikov, 2019, Chami et al., 2019], to image classification [Mathieu et al., 2019]. In these settings, the objects, and in most cases their relationships, are explicitly encoded in the data. However, 3D biomedical images consist of subvolumes that represent parts of an *implicit* hierarchical structure. In our case, for any single 3D voxel-grid, we wish to embed and infer the implicit relationships between all of its subvolumes without any supervision.

To embed our 3D images in hyperbolic space, we use a 3D hyperbolic variational autoencoder (VAE). For the decoder, we propose a gyroplane convolutional layer which maps from the latent space back to 3D images while respecting hyperbolic geometry. In addition to the VAE loss, we propose an essential self-supervised loss to capture the hierarchical structure present in the data. More specifically, we consider reconstruction of implicit hierarchies as a pretext task. Concretely, we add a triplet loss which encourages a child subvolume to be mapped close to its parent subvolume in hyperbolic space. To capture hierarchical relationships of varying granularity, we train on subvolumes sampled at multiple scales. Finally, for a specified scale, we cluster the subvolumes in latent space and produce a segmentation map.

We evaluate our model on datasets with different domains: synthetic datasets and a medical image dataset. We construct synthetic datasets where we generate structures at various scales and show that our model segments objects at multiple levels of hierarchy better than all prior unsupervised segmentation methods. We demonstrate performance gains ranging from 7% for the smallest objects to 32% for the largest objects. On the real-world medical image dataset (BraTS Brain Tumor Segmentation Challenge) [Menze et al., 2014, Bakas et al., 2017, 2018], we show that our method outperforms prior works by 19%, and even achieves comparable performance to semi-supervised methods although we do not use any labels.

## 2 Related Work

**Segmentation on 3D voxel data**    Many diverse biomedical images, ranging from MRI and CT scans to fluorescence microscopy, come in the form of 3D voxel-grids. Since 3D voxel-grids are dense, computer vision tasks such as supervised segmentation are commonly performed using deep learning architectures with 3D convolutional layers [Chen et al., 2016, Dou et al., 2017, Hesamian et al., 2019, Zheng et al., 2019]. However, due to the challenges of obtaining voxel-level annotations in 3D, there has been significant effort in finding semi-supervised approaches, including using labels only from several fully annotated 2D slices of an input volume [Çiçek et al., 2016], using a smaller set of segmentations with joint segmentation and registration [Xu and Niethammer, 2019], and using one segmented input in conjunction with other unlabelled data [Zhao et al., 2019].

Unsupervised approaches for 3D segmentation are useful not only for further reducing the manual annotation effort required, which is especially expensive for segmentation, but also for scientific discovery tasks where we would like to identify previously unknown structures for which annotations are impossible to produce. Moriya et al. [2018] extends to 3D data an iterative approach of feature learning followed by clustering [Yang et al., 2016]. Nalepa et al. [2020] uses a 3D convolutional autoencoder architecture and performs clustering of the latent representations. Another approach, [Dalca et al., 2018], uses a network pre-trained on manual segmentations from a separate dataset to perform unsupervised segmentation of 3D biomedical images. However, this limits applicability to areas where we already have a dataset with manual annotations and makes it unsuitable for unbiased scientific discovery. Gur et al. [2019] and Kitrungrotsakul et al. [2019] develop unsupervised methods

for 3D segmentation of vessel structures, but these are specialized and do not generalize to the segmentation of other structures, [Uzunova et al., 2019] utilizes knowledge of background patches with no patholoogy, and Baur et al. [2018] uses deep autoencoding models for unsupervised anomaly detection.

Another line of work performs unsupervised 2D segmentation, such as Ji et al. [2019] which proposes a mutual information objective for clustering, and Caron et al. [2018], which uses the clustered output of an encoder as pseudo-labels. While these methods can be applied to 2D slices of a 3D volume to perform 3D segmentation, they generally suffer limitations due to insufficient modeling of the 3D spatial information. None of the aforementioned approaches explicitly learn hierarchical structure of the data, which is the main focus of our work.

**Hyperbolic representations**   A recent line of work employs hyperbolic space to represent hierarchical structures, with the intuition that tree structures can be naturally embedded into continuous hyperbolic space [Nickel and Kiela, 2017]. These works utilize hyperbolic representations for a variety of tasks, including MNIST classification [Mathieu et al., 2019, Nagano et al., 2019, Ovinnikov, 2019], natural language processsing tasks such as embedding word taxonomies and entailment [Nickel and Kiela, 2017, Ganea et al., 2018], link prediction and node classification [Chami et al., 2019], and game playing [Nagano et al., 2019]. In most of these works, hierarchical structure is *explicitly* encoded in data. In contrast, we seek to capture *implicit* hierarchical structure arising from composition within 3D images.

Several architectures have been proposed in order to learn hyperbolic representations, including hyperbolic VAEs [Mathieu et al., 2019, Nagano et al., 2019, Ovinnikov, 2019], feed-forward and recurrent hyperbolic neural networks architectures [Ganea et al., 2018], and hyperbolic graph convolutional networks [Chami et al., 2019]. We extend the hyperbolic VAE framework to the task of learning hyperbolic representations from subvolumes of complex 3D images, and use this to perform unsupervised segmentation.

**Self-supervision**   Providing self-supervision by solving pretext tasks is one common approach for learning unsupervised visual representations. Pretext tasks leverage properties of the input data or prior knowledge as supervisory signals in order to learn better representations. Examples of pretext tasks include finding the relative position of two patches sampled from an image [Doersch et al., 2015], solving jigsaw puzzles [Noroozi and Favaro, 2016], and predicting pixel movements of videos in subsequent frames [Pathak et al., 2017]. In contrast, we propose the pretext task of reconstructing implicit hierarchy in 3D voxel-grid images, to learn effective hyperbolic representations for downstream segmentation.

## 3   Methods

In this section, we describe our approach for learning hyperbolic representations of subvolumes (3D patches) from 3D voxel-grid data. We propose a model that comprises a 3D convolutional variational autoencoder (VAE) with hyperbolic representation space and a gyroplane convolutional layer. We train our model with self-supervision through a novel hierarchical triplet loss and multi-patch sampling scheme. Then, we cluster the learned representations using hyperbolic $k$-means to produce 3D segmentations. In Section 3.1, we provide an overview of hyperbolic space. In Section 3.2, we describe our VAE framework with the proposed gyroplane convolutional layer and self-supervised hierarchical triplet loss. Finally, in Section 3.3, we discuss our approach of hyperbolic clustering for segmentation.

### 3.1   Hyperbolic formulation

**Hyperbolic space**   We embed subvolumes of 3D voxel-grid data in hyperbolic space, a non-Euclidean space with constant negative curvature. In negative curvature spaces, the area of a disc increases exponentially with the radius. We can think of this growth as analogous to the exponential increase of leaves at each level of a tree. Hence hyperbolic space can encode trees with arbitrarily low error [Sarkar, 2011] and can be considered as the continuous version of hierarchical structures. Unlike trees, hyperbolic space is smooth, permitting our use of deep learning on representations. For additional background on geometry and hyperbolic space, see the Appendix.

**Poincaré ball model of hyperbolic geometry**    In this work we use the Poincaré ball as our model of hyperbolic geometry. The Poincaré ball (of curvature $c = -1$) is the open ball of radius 1 centered at the origin equipped with the *metric tensor* $\mathfrak{g}_p = (\lambda_x)^2 \mathfrak{g}_e$, where the conformal factor $\lambda_x = \frac{2}{1-||x||^2}$ and $\mathfrak{g}_e$ is the Euclidean metric tensor (i.e., the Euclidean dot product). Formally, this makes the Poincaré ball a Riemannian manifold. For an introduction to Riemannian manifolds, see the Appendix. The distance $\mathbf{d_p}$ between points on the Poincaré ball is given by:

$$\mathbf{d_p}(x, y) = \cosh^{-1}\left(1 + 2\frac{||x - y||^2}{(1 - ||x||^2)(1 - ||y||^2)}\right) \tag{1}$$

We use the exponential and logarithm maps to map from Euclidean space to the Poincaré ball and vice versa. On the Poincaré ball, we note that the exponential and logarithm maps have the closed form expressions

$$\exp_z(v) = z \oplus \left(\tanh\left(\frac{\lambda_z ||v||}{2}\right)\frac{v}{||v||}\right) \tag{2}$$

$$\log_z(y) = \frac{2}{\lambda_z}\tanh^{-1}(|| - z \oplus y||)\frac{-z \oplus y}{|| - z \oplus y||} \tag{3}$$

where $\oplus$ denotes Mobius addition, which was introduced by Ungar [2001] as a way to define vector operations on hyperbolic space.

**Wrapped Normal Distribution**    The importance of the normal distribution in Euclidean space has led to many generalizations of the normal distribution to Riemannian manifolds. We use the wrapped normal distribution [Mathieu et al., 2019, Nagano et al., 2019], which can be defined on an arbitrary Riemannian manifold as the push-forward measure obtained by mapping the normal distribution in Euclidean space along the manifold's exponential map. On the Poincaré ball, the probability density function of the wrapped normal with mean $\mu$ and covariance $\Sigma$ is:

$$\mathcal{N}_P(z|\mu, \Sigma) = \mathcal{N}_E(\lambda_\mu(z)|0, \Sigma)\left(\frac{\mathbf{d_p}(\mu, z)}{\sinh(\mathbf{d_p}(\mu, z))}\right) \tag{4}$$

where the subscripts $P, E$ indicate distributions over the Poincaré ball and Euclidean space, respectively. We use the sampling and reparametrization scheme of Mathieu et al. [2019] in order to sample and train our VAE using the wrapped normal distribution.

## 3.2    Unsupervised hyperbolic representation learning

**3D hyperbolic variational autoencoder**    We propose a hyperbolic VAE that consists of a 3D convolutional encoder and decoder to handle 3D input. Our 3D convolutional encoder maps sampled subvolumes of the input into hyperbolic space and produces the parameters of the variational posterior. Our 3D convolutional decoder then reconstructs the 3D subvolumes from sampled latent hyperbolic representations. To ensure that both the encoder and decoder respect the geometry of the latent space, we follow Mathieu et al. [2019] and apply an exponential map to the output of the encoder, and use our novel gyroplane convolutional layer as the first layer of the decoder. We define the prior and variational posterior to be the wrapped normal distribution, which encourages our hierarchical representations to spread out on the Poincaré ball. Figure 1 illustrates an overview of our VAE framework.

Our variational autoencoder takes as input a patch of fixed size $m \times m \times m$. The model learns representations of subvolumes from the input ($X$ in Figure 1) that can subsequently be used to perform voxel-level segmentation of the whole volume. To learn hierarchical structure in 3D images, we train the VAE on 3D patches generated using a multi-scale sampling scheme that samples patches of size $r \times r \times r$, where size $r$ is randomly sampled and resized to $m$. Our method learns to embed each patch as part of a hierarchy in hyperbolic space.

**Gyroplane convolutional layer**    For learning better hyperbolic representations of 3D images, we introduce a gyroplane convolutional layer to effectively map from hyperbolic embedding space to Euclidean space. This allows us to keep the advantages of convolutional layers, such as locality, weight sharing, and translation equivariance. Our model's encoder output ($\mu$ in Figure 1) has a

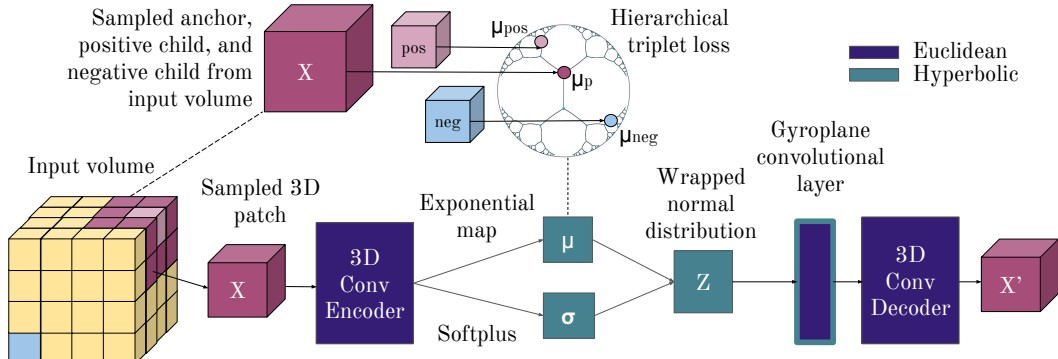

Figure 1: Our method learns hyperbolic representations of subvolumes of 3D voxel-grid data through a 3D hyperbolic VAE with a gyroplane convolutional layer. We enhance the VAE training objective with a self-supervised hierarchical triplet loss that facilitates learning implicit hierarchical structure within the VAE's hyperbolic latent space.

product manifold structure, since it is a Cartesian product of vectors in hyperbolic space. To map this to a 3D image in Euclidean space, we generalize the usual Euclidean convolutional layer by replacing the Euclidean affine transformation with an affine transformation on the manifold.

One way to define an affine transformation on the Poincaré ball is the gyroplane layer [Ganea et al., 2018]. The derivation of the gyroplane layer is motivated by the fact we can express a Euclidean affine transformation as: $\langle a, z - p \rangle = \text{sgn}(\langle a, z - p \rangle)||a||\mathbf{d_E}(z, H_{a,p})$ where $\mathbf{d_E}$ is Euclidean distance and $H_{a,p} = \{z \in \mathbb{R}^p | \langle a, z - p \rangle = 0\}$. $H_{a,p}$ is called the decision hyperplane. Ganea et al. [2018] defines the gyroplane layer $f_{a,p}$ from this formulation by replacing each component with its hyperbolic equivalent:

$$f_{a,p}(z) = \text{sgn}\left(\langle a, \log_p(z)\rangle_p\right)|a|_p\mathbf{d_p}(z, H_{a,p}) \tag{5}$$

where $H_{a,p}$ is the hyperbolic decision boundary $H_{a,p} = \{z \in \mathcal{B} | \langle a, \log_p(z)\rangle = 0\}$, and the distance to the hyperbolic decision boundary $\mathbf{d_p}(z, H_{a,p})$ is

$$\mathbf{d_p}(z, H_{a,p}) = \sinh^{-1}\left(\frac{2|\langle -p \oplus z, a\rangle|}{(1 - || - p \oplus z||^2)||a|}\right) \tag{6}$$

We can now define our gyroplane convolutional layer by generalizing the Euclidean affine transformation using the gyroplane layer. For simplicity, suppose $x$ is a 4D tensor containing elements of the Poincaré ball and our kernel size is $k \times k \times k$, with an odd $k$ value. Our gyroplane convolutional layer is defined as:

$$y_{r,s,t} = \sum_{\alpha=r-\lfloor k/2 \rfloor}^{r+\lfloor k/2 \rfloor} \sum_{\beta=s-\lfloor k/2 \rfloor}^{s+\lfloor k/2 \rfloor} \sum_{\gamma=t-\lfloor k/2 \rfloor}^{t+\lfloor k/2 \rfloor} f_{a,p}(x_{\alpha,\beta,\gamma}) \tag{7}$$

The gyroplane convolutional layer can be extended in the same way as Euclidean convolutional layers to incorporate even kernel size $k$, input and output channels, padding, stride, and dilation.

**Self-supervised hierarchical triplet loss**  As our model is trained on subvolumes of the 3D input, we cannot easily obtain the *implicit* hierarchical structure of the whole volume. To encode this structure in our model, we introduce self-supervision through the reconstruction of inferred hierarchy as a pretext task. This task encourages our learned representations on the Poincaré ball to reflect parent-child relationships of the input's implicit hierarchy.

Our self-supervision takes the form of a triplet loss that encourages hierarchically-related patches to have more similar representations. Since any two arbitrary patches may have some hierarchical relationship, we sample patches for our triplet loss to capture hierarchy in a tractable way. To sample 3D patches for our triplet loss, we first sample an anchor patch that acts as our parent patch (red

volume $X$ in Figure 1). We then sample the positive patch as a smaller subpatch that resides within the anchor patch (pink volume $pos$ in Figure 1). The anchor and positive patches form a parent-child relationship that we encourage to have closer representations in hyperbolic space, which has the interpretation as belonging to the same branch of the hierarchy ($\mu_p$ and $\mu_{pos}$ in Figure 1). The exponentially growing surface area near the edge of hyperbolic space allow this natural parent-child structure to form. We then sample a negative patch as a spatially distant patch (blue volume $neg$) that does not overlap with the anchor patch. The triplet loss encourages the the negative patch's representation to belong to a different branch of the hierarchy ($\mu_p$ and $\mu_{neg}$).

Our choice of positive and negative patches is motivated by the compositional hierarchy of 3D volumes. The hierarchical triplet loss encourages the anchor patch and a sub-patch (parent and positive child) to have similar representations, while encouraging the anchor patch and a distant patch (parent and negative child) to have dissimilar representations. We use this implicit hierarchy reconstruction as a pretext task to encourage learning relationships between nested objects in 3D biomedical images. Our multi-patch sampling scheme and triplet loss formulation allows representations to encode *implicit* structure in hyperbolic space.

Our hierarchical triplet loss can be formulated with any dissimilarity measure $\mathbf{d}$ between the encoder outputs $\mu$ (see Figure 1) of the anchor $\mu_p$, positive child $\mu_{pos}$, and negative child $\mu_{neg}$. For our model, we take $\mathbf{d}$ to be the Poincaré ball distance $\mathbf{d_p}$ and define our triplet loss with margin $\alpha$ as:

$$L_{triplet}(\mu_p, \mu_{pos}, \mu_{neg}) := \max(0, \mathbf{d_p}(\mu_p, \mu_{pos}) - \mathbf{d_p}(\mu_p, \mu_{neg}) + \alpha) \tag{8}$$

This formulation can be extended to any metric space by taking the dissimilarity measure $\mathbf{d}$ to be the space's metric. In particular, for our ablations using an Euclidean latent space, we take the dissimilarity measure $\mathbf{d}$ to be the Euclidean distance.

**Optimization** We optimize a loss function that can be decomposed as the standard evidence lower bound (ELBO) loss for variational autoencoders and our hierarchical triplet loss that encourages the learning of structure in hyperbolic space. Mathieu et al. [2019] generalized the ELBO loss to Riemannian manifold latent spaces as

$$L_{ELBO} := \int_{\mathcal{M}} \log\left(\frac{p_\theta(x|z)p(z)}{q_\phi(z|x)}\right) q_\phi(z|x)d\mathcal{M}(z) \leq \log p(x) \tag{9}$$

where $d\mathcal{M}(z) = \sqrt{|G(z)|}dz$ is the measure induced on the manifold by the Riemannian metric $G(z)$ (see Appendix). Our total loss is then formulated as

$$L_{total} = L_{ELBO} + \beta L_{triplet} \tag{10}$$

where $\beta$ is a hyperparameter that controls the strength of the hierarchical triplet loss.

### 3.3 Segmentation by clustering representations

**Hyperbolic clustering** In 3D segmentation, we seek to assign each voxel $v$ a segmentation label $s_v \in \{1, \ldots, n\}$, where $n$ is the number of segmentation classes. We perform segmentation by clustering the representations of patches centered at each voxel. We first use our trained VAE encoder to generate latent representations $\mu_v$ for each voxel $v$. We do this by taking a patch of fixed size $p \times p \times p$ centered at $v$, upsampling or downsampling it to the encoder input size $m \times m \times m$, and then encoding the patch to retrieve $\mu_v$. We then cluster the $\mu_v$ into $n$ clusters, and produce a segmentation by assigning each $v$ the cluster label of $\mu_v$. We perform clustering through a $k$-means algorithm that respects hyperbolic geometry, which we derive by replacing the Euclidean centroid and distance computations of classical $k$-means with their counterparts in Riemannian geometry, the Fréchet mean and manifold distance. We calculate the Fréchet mean using the algorithm of Lou et al. [2020].

## 4 Experiments

We evaluate our method quantitatively on both synthetic 3D datasets simulating biological image data as well as the real-world Brain Tumor Segmentation (BraTS) tumor segmentation dataset. Our biologically-inspired synthetic datasets allows quantitative evaluation of segmentation at multiple levels of hierarchy, while the BraTS dataset is a well-known benchmark for 3D MRI segmentation.

**Implementation details**  For all models, the encoder of our variational autoencoder is comprised of four 3D convolutional layers with kernel size 5 of increasing filter depth $\{16, 32, 64, 128\}$. The decoder has the same structure, except with decreasing filter depth and a gyroplane convolutional layer as the initial layer. We use $\beta = 1e3$ as the weighting factor between $L_{\mathrm{ELBO}}$ and $L_{\mathrm{triplet}}$ and $\alpha = 0.2$ as the triplet margin. In all experiments, we fix the representation dimension to be $d = 2$, and show latent dimension ablations in the Appendix. We train our model using the Adam optimizer [Kingma and Ba, 2014]. For inference, we obtain the latent representations of $5 \times 5 \times 5$ patches densely across the full volume, and then perform hyperbolic $k$-means clustering, where the number of clusters $k$ is a hyperparameter that controls the granularity of the segmentation. For quantitative evaluation, we then use the Hungarian algorithm [Kuhn, 1955] to match each predicted segmentation class with a corresponding ground truth label.

We utilize two anchor patch sampling schemes, one for input of smaller sizes and one for larger sizes. In both schemes, for a given 3D volume, we sample $i$ patch centers $v_i$ uniformly with patch size $r$, and upsampling or downsampling to size $m \times m \times m$. In the sampling scheme for smaller inputs, the patch size $r$ is sampled uniformly, whereas in the sampling scheme for larger inputs, $r$ is sampled log-uniformly. This scheme is motivated by the following observations: for larger patches, a small change in $r$ is less likely to correspond to significant semantic difference, and inherent structure causes the different levels of hierarchy to naturally follow a log scale. For training on the synthetic dataset, we sample 3D volume sizes uniformly, and for BraTS we sample using the log scale.

For every sample of an anchor patch of size $r \times r \times r$, we generate a positive child patch as a smaller patch of the anchor patch as follows: the positive child patch is a subvolume within the anchor patch with size $r_{\mathrm{child}} \times r_{\mathrm{child}} \times r_{\mathrm{child}}$, where $r_{\mathrm{child}} \sim \mathcal{U}(\ell_{\min}, r - r_{\mathrm{gap}})$, and $r_{\mathrm{gap}}$ is a hyperparameter representing the gap in size between the anchor size and the child size. The negative child is a patch of size $r_{\mathrm{child}} \times r_{\mathrm{child}} \times r_{\mathrm{child}}$ that does not overlap with the anchor patch.

### 4.1   Biologically-inspired synthetic dataset

Since we want to evaluate segmentation performance at multiple levels of hierarchy and most 3D datasets do not have the necessary annotation, we first generate a synthetic dataset. This dataset enables a more thorough evaluation of the effectiveness of our model for unsupervised 3D segmentation. Our synthetic dataset is inspired by cryo-ET images of cells. Each volume in our synthetic dataset contains multiple levels of hierarchy with the objects at each level differentiated by texture, size, and shape. Figure 2 shows an example input volume with sampled slices shown. Our dataset consists of 120 total volumes, which we split into 80 training, 20 validation, and 20 test examples. Each synthetic volume has size $50 \times 50 \times 50$. Additional information on the synthetic dataset generation process as well as a more difficult version of the dataset, where the boundaries of each shape are perturbed, is described and benchmarked in Appendix A.5.

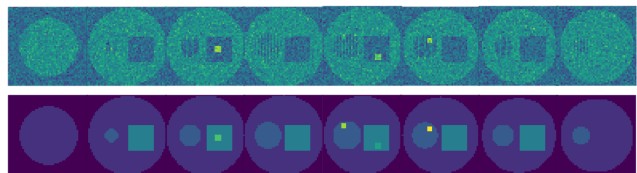

Figure 2: Sampled 2D slices from a 3D volume in our biologically-inspired synthetic dataset. The top row showcases the raw input data, and the bottom row showcases the ground truth segmentation.

To demonstrate segmentation performance on objects at different scales, we evaluate on the three levels of hierarchy defined above and use the average class DICE score to compare segmentation performance. Since our model is unsupervised, we assign segmentation classes to ground truth labels using the Hungarian algorithm. See results in Table 1 and Table 2. We also show results on a more challenging irregular synthetic dataset in Appendix A.5.

**Comparison with prior approaches**  Table 1 shows quantitative comparison of our method with prior state-of-the-art 3D unsupervised and 2D unsupervised (which we extend to 3D) models. In addition, we also compare our method to prior semi-supervised work, as unsupervised 3D segmentation is a relatively unexplored field, and we provide baselines with different levels of supervision for

Table 1: Comparison with prior approaches on synthetic dataset

|  | DICE *Level 1* | DICE *Level 2* | DICE *Level 3* | SUPERVISION TYPE |
|---|---|---|---|---|
| ÇIÇEK ET AL. [2016] | 0.968 | 0.829 | 0.668 | 3D SEMI-SUPERVISED |
| ZHAO ET AL. [2019] | 0.989 | 0.655 | 0.357 | 3D SEMI-SUPERVISED |
| NALEPA ET AL. [2020] | 0.530 | 0.276 | 0.112 | 3D UNSUPERVISED |
| JI ET AL. [2019] | 0.589 | 0.291 | 0.150 | 2D TO 3D UNSUPERVISED |
| MORIYA ET AL. [2018] | 0.628 | 0.311 | 0.141 | 3D UNSUPERVISED |
| **OURS** | **0.952** | **0.541** | **0.216** | 3D UNSUPERVISED |

Table 2: Ablation studies on synthetic dataset

| LATENT SPACE | CONFIGURATION | DICE *Level 1* | DICE *Level 2* | DICE *Level 3* |
|---|---|---|---|---|
| EUCLIDEAN | BASE | 0.784 | 0.322 | 0.109 |
|  | TRIPLET | 0.761 | 0.342 | 0.153 |
| HYPERBOLIC | BASE | 0.832 | 0.352 | 0.135 |
|  | GYROCONV | 0.905 | 0.473 | 0.204 |
|  | TRIPLET | 0.945 | 0.534 | **0.222** |
|  | GYROCONV & TRIPLET | **0.952** | **0.540** | 0.216 |

additional reference. Çiçek et al. [2016] was trained with $2\%$ of the ground truth slices in each of the $xy$, $yz$, and $xz$ planes, and Zhao et al. [2019] was trained with one fully annotated atlas, which can both still be expensive given the large size of many 3D biomedical images. Ji et al. [2019] was implemented using the authors' original code and extrapolated to 3D by applying the method to each slice. For Nalepa et al. [2020] and Moriya et al. [2018], we re-implemented their methods as the original code was unavailable. Our model performs significantly better than all unsupervised prior work at all levels of hierarchy. We also perform comparably to the semi-supervised approach of Zhao et al. [2019], despite not using any labels.

**Ablation**   Table 2 presents ablation studies on the hierarchical synthetic dataset comparing our contributions: Euclidean vs. hyperbolic representations, the addition of our gyroplane convolutional layer, and the addition of our hierarchical triplet loss. The base Euclidean configuration is the 3D convolutional VAE with Euclidean latent space, no gyroplane convolutional layer, and trained with just the ELBO loss. The triplet Euclidean configuration adds the hierarchical triplet loss to the base Euclidean configuration. The base hyperbolic configuration is the same as the base Euclidean configuration except with hyperbolic latent space. The triplet configuration is the hyperbolic analogue of the Euclidean triplet configuration, and gyroconv configurations have the addition of the gyroplane convolutional layer.

Hyperbolic representations outperform their Euclidean counterparts in all experiments. We attribute this to the more efficient and better organization of hyperbolic representations. When we introduce our self-supervised triplet loss, performance improves significantly for our hyperbolic models. Performance for our Euclidean model does not improve as much, likely due to information loss in representing hierarchical input. Introducing the gyroplane convolutional layer shows clear improvement over the base hyperbolic model, which demonstrates the benefit of having a convolutional layer that respects the geometry of the latent space. The combination of the triplet loss and gyroplane convolutional layer exhibits the most gain over the base hyperbolic model, with smaller gains over the model with just the added triplet loss. We show the importance of our hierarchical self-supervision for learning effective representations that capture implicit hierarchical structure.

## 4.2   Brain Tumor Segmentation challenge dataset

The BraTS 2019 dataset is a public, well-established benchmark dataset containing 3D MRI scans of brain tumors and voxel-level ground truth annotations of tumor segmentation masks [Menze et al., 2014, Bakas et al., 2017, 2018]. The scans are of dimension $200 \times 200 \times 155$ and have four modalities; we use the FLAIR modality, which is the most commonly used one-modality input. We

Table 3: Table shows comparison on BraTS 2019 dataset. Figure shows a qualitative example where top left image is a slice from a 3D test volume, and the three other images show segmentations with $2, 3, 4$ numbers of clustering centroids respectively, illustrating multiple levels of hierarchy learned.

| BRATS DATASET | DICE WT | ALGORITHM TYPE |
|---|---|---|
| SOTA [JIANG ET AL., 2019] | 0.888 | 3D FULLY-SUP. |
| ZHAO ET AL. [2019] | 0.648 | 3D SEMI-SUP. |
| ÇIÇEK ET AL. [2016] | 0.760 | 3D SEMI-SUP. |
| JI ET AL. [2019] | 0.211 | 2D-TO-3D UNSUP. |
| MORIYA ET AL. [2018] | 0.425 | 3D UNSUP. |
| NALEPA ET AL. [2020] | 0.495 | 3D UNSUP. |
| **OURS** | **0.684** | 3D UNSUP. |

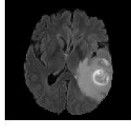 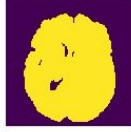
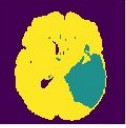 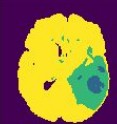

use the same evaluation metric as in the BraTS challenge, and compare DICE score on whole tumor (WT) segmentation, which is detectable solely from FLAIR, as well as average and 95 percentile Hausdorff distance for the competing unsupervised methods (see Table 4). There are 259 high grade glioma (HGG) labelled training examples, which we split into 180 train, 39 validation, and 40 test examples. We do not use the official validation or test sets because the ground truth annotations for these sets are not publicly available.

Table 3 shows the DICE score comparison of our results against prior work. For fair comparison, all baselines are trained with only the FLAIR modality. The only exception is the current state-of-the-art fully-supervised result [Jiang et al., 2019] in Table 3, which uses all 4 modalities. We show this for reference as an upper bound; the reported number is trained on the full train set and evaluated on the BraTS test set.

Our best model performs significantly better than all prior unsupervised methods, and in addition outperforms one 3D semi-supervised model. This demonstrates the ability of our self-

Table 4: Comparison of our method against prior unsupervised work in Hausdorff distance. (Lower is better.)

| | AVERAGE | 95% |
|---|---|---|
| MORIYA ET AL. [2018] | 118.144 | 170.434 |
| JI ET AL. [2019] | 96.865 | 114.400 |
| NALEPA ET AL. [2020] | 87.704 | 110.803 |
| **OURS** | **77.940** | **97.641** |

supervised hyperbolic representations to effectively capture the hierarchical structure in individual brain scans. We use a granular segmentation with three clusters for quantitative evaluation in order to capture the tumor, brain, and background, then use the Hungarian algorithm for assignment. In Table 4, we demonstrate that our model also outperforms all prior methods on average and 95 percentile Hausdorff metrics. In addition, we also show qualitative results for our model (see Figure 3), which include byproduct segmentations from the same model with different numbers of clusters specified, showcasing additionally discovered features in the scan that could be clinically useful.

## 5 Conclusion

We propose a method for learning hyperbolic representations of 3D voxel-grid images that captures the *implicit* hierarchical structure in biomedical data, and show that these representations are well suited for the task of unsupervised 3D segmentation. We conduct our representation learning through a hyperbolic 3D convolutional VAE with a novel gyroplane convolutional layer that respects hyperbolic geometry. We then enhance the VAE training objective with a self-supervised hierarchical triplet loss that learns *implicit* hierarchical structure within the VAE's hyperbolic latent space as a pretext task. We demonstrate that hyperbolic clustering of learned voxel-level representations can be used to achieve state-of-the-art unsupervised 3D segmentation on synthetic hierarchical datasets and the real-world BraTS dataset.

**Acknowledgments.** We thank support from the Chan-Zuckerberg Initiative (Neurodegeneration Challenge Network Collaborative Pairs grant DAF2021-221728 to J.H., J.G., G.W., W.C., and S.Y.) and National Institutes of Health (grant number P01NS092525 to W.C.).

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
