# A   Appendix

## A.1   Broader impact

Our work introduces a general method for unsupervised 3D segmentation that can be used for any 3D voxel-grid data. This line of work is especially useful for analyzing biomedical data, as many different types of biomedical data are in volumetric form and lack the ground truth annotations required for fully- or semi-supervised segmentation. For example, we may wish to study diseased tissue but do not have sufficient understanding to ensure that unexplored features of interests are labelled in training data. We illustrate the potential of our proposed approach for scientific discovery applications using our example of cryo-ET data in the Appendix. The discovered features can now be analyzed for their chemical identities and functions, in diseased vs. healthy cells. Similarly, unsupervised discovery of substructures can also enable richer analysis of other types of biomedical data such as CT and MRI scans. Potential negative societal impact of our work could arise from malicious intent in extracting information from certain types of 3D voxel-grid data for ill use, such as data mining from 3D scenes of sensitive domains without consent, which our method facilitates easily without labels. However, we hope our work is utilized to enable new downstream applications primarily from real-world 3D biomedical images, which are one of the most common types of 3D voxel-grid data.

## A.2   Riemannian manifolds

In this section, we give a more complete introduction to Riemannian manifolds, of which hyperbolic space is an example. Riemannian manifolds are spaces that locally resemble Euclidean space. To define this mathematically, we first introduce a *manifold* as a set of points $\mathcal{M}$ that locally resembles the Euclidean space $\mathbb{R}^n$. Associated with each point $\mathbf{x} \in \mathcal{M}$ is a vector space called the *tangent space* at $\mathbf{x}$, denoted $\mathcal{T}_{\mathbf{x}}\mathcal{M}$, which is the space of all directions a curve on the manifold $\mathcal{M}$ can tangentially pass through point $\mathbf{x}$. A metric tensor $\mathfrak{g}$ defines an inner product $\mathfrak{g}_{\mathbf{x}}$ on every tangent space, and a *Riemannian manifold* is a manifold $\mathcal{M}$ together with a metric tensor $\mathfrak{g}$. For each tangent sapce $\mathcal{T}_x\mathcal{M}$, the metric tensor has *matrix representation* $G$ defined as $\mathfrak{g}_{\mathbf{x}}(u, v) = u^T G(\mathbf{x})v$.

Distance on a Riemannian manifold as can defined as the following. Let $\gamma : [a, b] \rightarrow \mathcal{M}$ be a curve on the manifold $\mathcal{M}$. The *length* of $\gamma$ is defined to be $\int_a^b |\gamma'(t)|_{\gamma(t)}dt$ and denoted $L(\gamma)$. The *distance* between any two points $\mathbf{x}, \mathbf{y}$ on the manifold is defined as $d_{\mathcal{M}}(\mathbf{x}, \mathbf{y}) = \inf L(\gamma)$, where the inf is taken over all curves $\gamma$ that begin at $\mathbf{x}$ and end at $\mathbf{y}$. This distance makes $\mathcal{M}$ a metric space.

The *exponential map* $\exp_{\mathbf{x}}(v) : \mathcal{T}_{\mathbf{x}}\mathcal{M} \rightarrow \mathcal{M}$ is a useful way to map vectors from the (Euclidean) tangent space to the manifold. The exponential map is defined as $\exp_{\mathbf{x}}(v) = \gamma(1)$, where $\gamma$ is the unique geodesic, the shortest possible curve between two points, starting at $\mathbf{x}$ with starting direction $v$. Intuitively, one can think of the exponential map as telling us how to travel one step starting from a point $\mathbf{x}$ on the manifold in the $v$ direction. The logarithmic map $\log_v(x) : \mathcal{M} \rightarrow \mathcal{T}_{\mathbf{x}}\mathcal{M}$ is the inverse of the exponential map, and maps vectors back to Euclidean space.

## A.3   Gyrovector operations in the Poincaré Ball

Gyrovector operations were first introduced by Ungar [2008] to generalize the Euclidean theory of vector spaces to hyperbolic space. Mobius addition is the Poincaré ball analogue of vector addition in Euclidean spaces. The closed-form expression for Mobius addition on the Poincaré ball with negative curvature $c$ is Mathieu et al. [2019]:

$$z \oplus_c y = \frac{(1 + 2c\langle z, y\rangle + c||y||^2)z + (1 - c||z||^2)y}{1 + 2c\langle z, y\rangle + c^2||z||^2||y||^2} \tag{1}$$

As one might anticipate, when $c = 0$ we recover Euclidean vector addition. Additionally, the analogue of Euclidean vector subtraction is Mobius subtraction, which is defined as $x \ominus_c y = x \oplus_c (-y)$, and the analogue of Euclidean scalar multiplication is Mobius scalar multiplication, which can be defined for a scalar $r$ as [Ganea et al., 2018]:

$$r \otimes_c x = \frac{1}{\sqrt{c}} \tanh(r \tanh^{-1}(\sqrt{c}||x||)) \frac{x}{||x||} \tag{2}$$

where we also recover Euclidean scalar multiplication when $c = 0$. In this paper, we only consider the Poincaré ball with fixed constant negative curvature $c = 1$, which allows us to drop the dependence on $c$.

Table 1: Ablation study of latent space dimension for Euclidean and Hyperbolic models on the synthetic dataset. Dice scores for all three levels are reported.

| LATENT SPACE | DICE *Level* | D=2 | D=3 | D=5 | D=8 | D=16 |
|---|---|---|---|---|---|---|
| HYPERBOLIC | *Level 1* | 0.952 | 0.959 | 0.956 | 0.942 | 0.954 |
| | *Level 2* | 0.541 | 0.538 | 0.550 | 0.529 | 0.541 |
| | *Level 3* | 0.216 | 0.213 | 0.219 | 0.226 | 0.228 |
| EUCLIDEAN | *Level 1* | 0.761 | 0.838 | 0.847 | 0.871 | 0.872 |
| | *Level 2* | 0.342 | 0.362 | 0.378 | 0.481 | 0.495 |
| | *Level 3* | 0.153 | 0.176 | 0.165 | 0.225 | 0.228 |

## A.4 Latent dimension ablation

For all sections in our paper, our experiments were all run with latent dimension of 2. To show the effect of higher latent space dimensions, we report an ablation study for both hyperbolic and Euclidean representations (See Table 1). As expected, for our Euclidean latent space model, performance increases with dimension. However, our hyperbolic model still outperforms the Euclidean model at all tested dimensions, and shows that we can embed representations efficiently at lower dimensions.

## A.5 Biologically-inspired synthetic dataset and the irregular variant

Each 3D image of our biologically-inspired synthetic dataset consists of three levels of hierarchy. The first level of hierarchy (*Level 1*) has a noisy background and an outer sphere of radius $r \sim \mathcal{N}(25, 1)$. Using a cell analogy, this represents the entire cell. The second level (*Level 2*) consists of spheres ("vesicles") and cuboids ("mitochondria"). Their sizes are randomly sampled with radius of $r \sim \mathcal{N}(8, 0.5)$ and with side length of $s \sim 2 \cdot \mathcal{N}(8, 0.5)$, respectively. In the third level (*Level 3*) we introduce small spheres and cuboids ("protein aggregates") in the vesicle spheres and mitochondria cuboids respectively. The *Level 3* proteins have a radius of $r \sim \mathcal{N}(2, 0.2)$ and side length of $s \sim 2 \cdot \mathcal{N}(3, 0.15)$, respectively. The location of each object is sampled randomly, subject to the restriction that objects in Level $i + 1$ are entirely contained within an object in Level $i$.

Each instance of a shape with a particular size is also given its own unique texture to mimic the different organelles of the cell. The color of each object is chosen randomly, according to a standard normal distribution. We also apply pink noise with magnitude $m = 0.25$ to the volume as it is commonly seen in biological data.

We generate an additional synthetic dataset with irregular shapes for evaluating datasets with large variance in characteristics across levels of hierarchy. This dataset was created through applying smooth noise to the boundaries of each shape. Specifically, we generate noise by first sampling random points in our voxel-grid and random values according to a Gaussian distribution, and interpolating to retrieve smooth noise. We then use this smooth noise function to perturb the points that fall within the interior of the three largest shapes. See an example of the dataset in Figure 1.

We demonstrate our method's performance in comparison to prior work on the aforementioned irregular dataset in Table 2, and an ablation study applied on the same irregular dataset in Table 3.

We note that in Table 2, our proposed method outperforms prior work significantly on this irregular dataset, following our observations from our unperturbed synthetic dataset. We can see that while most methods show slight decrease in performance, our approach still shows state-of-the-art performance compared to prior unsupervised segmentation work across all hierarchical levels.

For ablations on the irregular synthetic dataset in Table 3, we find that our best models with hyperbolic latent space reliably outperform models with Euclidean latent space, as with our unperturbed synthetic dataset. Both Euclidean and hyperbolic base models have much lower performance on the irregular dataset compared to the unperturbed dataset, due to the challenges that the irregular dataset brings, for example, needing to recognize noisy instances of irregular shape as the same class. However, we demonstrate that the gyroplane convolutional layer and hierarchical triplet loss are both effective ways to improve performance on the base hyperbolic configuration. The inclusion of both of our contributions allows for significant performance gain across hierarchical levels, such that the results

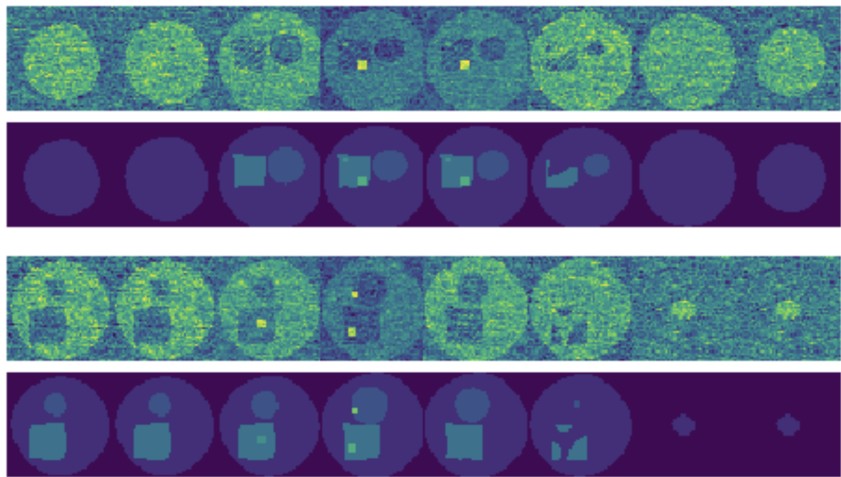

Figure 1: Sampled 2D slices from two examples of 3D volumes in our irregular biologically-inspired synthetic dataset, showing large variance in shapes across input. For each 3D volume example, the top row showcases the raw input data, and the bottom row showcases the ground truth segmentation.

are comparable to that of the unperturbed dataset, even with a 23% difference in *Level 1* base hyperbolic performance.

Table 2: Comparison with prior approaches on irregular synthetic dataset

|  | Dice *Level 1* | Dice *Level 2* | Dice *Level 3* | Supervision type |
| --- | --- | --- | --- | --- |
| Çiçek et al. [2016] | 0.970 | 0.825 | 0.601 | 3D Semi-supervised |
| Zhao et al. [2019] | 0.978 | 0.641 | 0.333 | 3D Semi-supervised |
| Nalepa et al. [2020] | 0.559 | 0.259 | 0.138 | 3D Unsupervised |
| Ji et al. [2019] | 0.527 | 0.280 | 0.144 | 2D to 3D Unsupervised |
| Moriya et al. [2018] | 0.525 | 0.232 | 0.094 | 3D Unsupervised |
| **Ours** | **0.953** | **0.488** | **0.199** | 3D Unsupervised |

Table 3: Ablation studies on irregular synthetic dataset

| Latent Space | Configuration | Dice *Level 1* | Dice *Level 2* | Dice *Level 3* |
| --- | --- | --- | --- | --- |
| Euclidean | Base | 0.581 | 0.230 | 0.122 |
|  | Triplet | 0.823 | 0.392 | 0.175 |
| Hyperbolic | Base | 0.607 | 0.284 | 0.158 |
|  | GyroConv | 0.812 | 0.401 | 0.197 |
|  | Triplet | 0.947 | **0.491** | 0.192 |
|  | GyroConv & Triplet | **0.953** | 0.488 | **0.199** |

### A.6 BraTS ablations, error bars, and Hausdorff distance

We conduct an ablation study on the BraTS dataset, with each of our added components with error bars over 4 independent runs. Results are shown in Table 4. We can see that our best hyperbolic model outperforms our best Euclidean model significantly. The addition of the triplet loss improved both Euclidean and hyperbolic models, while the hyperbolic models see more performance gain. Our gyroplane convolutional layer also improves performance, while both of our additions jointly improve upon our Hyperbolic baseline, showing the benefit of these components to learning effective representations.

Table 4: Ablation study for BraTS dataset. We report the mean and standard deviation of DICE scores for 4 independent runs.

| Latent Space | Configuration | Dice |
|---|---|---|
| Euclidean | Base | $0.388 \pm 0.022$ |
| | Triplet | $0.517 \pm 0.050$ |
| Hyperbolic | Base | $0.414 \pm 0.017$ |
| | GyroConv | $0.539 \pm 0.014$ |
| | Triplet | $0.610 \pm 0.028$ |
| | GyroConv & Triplet | $\mathbf{0.692} \pm 0.009$ |

We include the average and 95 percentile Hausdorff distance as complementary evaluation metrics on the BraTS dataset for comparison to prior unsupervised works in the main text. We describe the calculation below.

We use Hausdorff distance to evaluate the worst-case performance of our model. For two sets of points $A, B$, the directed Hausdorff distance from $A$ to $B$ is defined as

$$h(A, B) = \max_{a \in A} \left\{ \min_{b \in B} \mathbf{d}(a, b) \right\} \tag{3}$$

where $\mathbf{d}$ is any distance function. We will take $\mathbf{d}$ to be the Euclidean distance. The Hausdorff distance is then defined to be

$$H(A, B) = \max \left\{ h(A, B), h(B, A) \right\} \tag{4}$$

The official BraTS evaluation uses 95 percentile Hausdorff distance as measure of model robustness [Bakas et al., 2018].

The BraTS dataset is licensed under Creative Commons Attribution.

## A.7 DICE score

We use DICE score to quantitatively evaluate segmentation performance on all datasets. The DICE score is defined as the following:

$$DICE = \frac{2TP}{2TP + FN + FP} \tag{5}$$

where $TP$ is the number of true positives, $FN$ is the number of false negatives, and $FP$ is the number of false positives. For our synthetic dataset, we first assign predicted classes to ground truth labels using the Hungarian algorithm Kuhn [1955], then evaluate using the average class DICE score. For the BraTS dataset Menze et al. [2014], Bakas et al. [2017, 2018], we evaluate DICE of the whole tumor segmentation following official evaluation guidelines.

## A.8 Qualitative results in electron tomography

We show a real-world example where unsupervised segmentation of new biological organelles is important. Cryogenic electron tomography (cryo-ET) is a technique that images cells at cryogenic temperatures with a beam of electrons. The value of each voxel is the density at that location, and is created through reconstruction from tilt slices of $\pm 60$ degrees from electron tomography. Cryo-ET images are a rich source of biological data, capturing many unknown subcellular objects that we would like to identify and understand.

We train our model on three $512 \times 512 \times 250$ cryo-ET tomograms of cells collected from a research laboratory, and run inference on a fourth tomogram. Figure 2 shows segmentations produced by our model on a mitochondria from the evaluation tomogram, using our proposed hyperbolic model vs. Euclidean model, and at a coarse and finer level of granularity. Unlike the Euclidean approach, our hyperbolic approach discovers a fine-grained class corresponding to small features on the mitochondria, which may be macromolecular aggregates. We can now investigate the discovered features for their biological identities and functions, leading to greater scientific understanding.

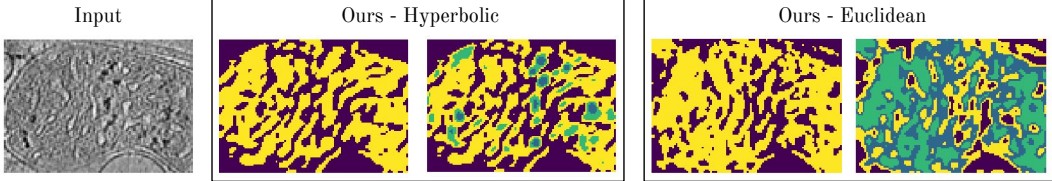

Figure 2: Leftmost image is a partial slice from a 3D cryo-ET image. The features of interest to be segmented are the dark densities with irregular shapes and sizes. The middle box shows segmentation from our best hyperbolic model, the rightmost box shows segmentation from our best Euclidean model. The segmentations in each box correspond to clustering using 2 vs. 4 classes.

## A.9  Hyperparameters

We use a single set of hyperparameters on all of our evaluation datasets, and these hyperparameters are not tuned on any of the evaluation datasets. In order to obtain a reasonable set of hyperparameters, we created a completely separate synthetic dataset on which we trained models and tuned hyperparameters. This synthetic dataset was created in a similar manner to our synthetic dataset; however, we designed it to have different and fewer objects, simpler nesting structure, no noise, and fewer textures. The application of this single set of hyperparameters to our evaluation datasets — our synthetic dataset, the BraTS dataset, and the cryogenic electron tomography dataset, demonstrates the robustness of our approach.

With the separate synthetic dataset that we used for choosing hyperparameters, we tuned over a range of values using its validation set. This includes weight of triplet loss in the range of $\beta = \{10^{-2}, 10^{-1}, 1, 10^{1}, 10^{2}, 10^{3}, 10^{4}, 10^{5}\}$, patch size for inference $p = \{5, 10, 15, 20, 40\}$, and number of epochs $e = \{3, 5, 8, 10, 12, 15\}$. We then used optimal hyperparameters $\beta = 10^{3}$, $p = 5$, and $e = 8$ for all experiments in our evaluation datasets. We used the Adam optimizer, w/ learning rate 1e-4, $\beta_1 = 0.9$, $\beta_2 = 0.999$. Training time of our model is between 5 to 8 hrs on a Titan RTX GPU.

## A.10  Reproducibility of prior work

Where available, we have used the authors' original code to generate the unsupervised baselines for the prior work comparisons. To sanity-check the code we used, we re-ran original experiments from the baseline paper. For [Ji et al., 2019], we re-ran their Potsdam-3 experiment for unsupervised 2D segmentation, and were able to reproduce the result from their paper to within approximately 1%. For [Moriya et al., 2018], neither the original code nor the original dataset are publicly available, making reproducibility impossible to check, however we used the code base which their method was based on to implement their work. For [Nalepa et al., 2020], the original code is unavailable as well, and we have adapted their method to our architecture in order to ensure a fair comparison.