# OpenReview forum: "Capturing implicit hierarchical structure in 3D biomedical images with self-supervised hyperbolic representations"
_NeurIPS.cc/2021/Conference — NeurIPS 2021 Poster_

### Official Review · Reviewer_oTyp · 2021-07-12

**Rating:** 4
**Confidence:** 4

**Summary:**

This paper proposes a  VAE architecture with a hyperbolic latent space called 3D hyperbolic VAE. The proposed method is evaluated on synthetic datasets and a medical dataset, showing its effectiveness for the unsupervised 3D segmentation task.

**Limitations And Societal Impact:**

There is no evident limitation in the proposed method.

**Main Review:**

**Originality:**  The novelty is limited, as the proposed method is a combination of exsiting techniques. However, I appreciate that this paper tries to solve a difficult task, i.e., unsupervised segmentation of 3D voxel-grid biomedical images.

**Quality:**  (1) I would suggest authors to use more evaluation criterions. In the 3D biomedical segmentation task, Jaccord score,  95% Hausdorff Distance (95HD), and  Average Surface Distance (ASD) are  also usually used. Providing results in terms of these criterions can better evaluate the proposed method.

(2) Table 3 shows the comparision with previous state-of-the-art methods on BraTS 2019. But there are not many methods on the unsupervised segmentation task. I think authors should consider more methods for comparision, even some frameworks in the computer vision field for the unsupervised segmentation of 3D data (e.g., videos). Plus, more real medical datasets can be used for evaluation.

**Clarity:**  The paper is clearly written and easy to follow.

Overall, I am concern about the novelty of the paper, and authors may need to provide more experimental results in terms of different evaluation indicators and more real datasets. So, I vote for rejection, but the score might be increased based on the authors' response and the discussion.

**Time Spent Reviewing:**

8

---

> ### Author Response · Authors · 2021-08-10
> **Response to Reviewer oTyp**
>
> We thank the reviewer for pointing out that we try to “solve a difficult task, i.e., unsupervised segmentation of 3D voxel-grid biomedical images” and that our “paper is clearly written and easy to follow.” We address specific reviewer questions below.
>
> Re: More evaluation metrics
>
> We thank the reviewer for the recommendation and agree that additional evaluation criterions would be helpful. In our appendix (A.6 BraTS ablations, error bars, and Hausdorff distance), we include 95% Hausdorff distance (95HD) and average Hausdorff distance to measure worst case and average distance respectively. The Jaccard score is a deterministic function of the DICE score and can be approximated from the DICE score. We would like to clarify that the requested criteria are either in our paper and appendix, or are variations of our metrics that can be directly computed what we have reported. We will add these variations of evaluation metrics in our final paper for completeness.
>
> Re: More unsupervised methods for comparison
>
> As the field of unsupervised segmentation is still relatively unexplored, and there are few works that tackle unsupervised 3D semantic segmentation of voxel grids (as compared to point clouds); we have included comparisons to prior semi-supervised works as well for completeness. Unsupervised segmentation of video data is not easily adapted, as many of the state-of-the-art methods tend to mainly consider the task of salient object segmentation, which only produces binary masks [1, 6], require the use of extra annotated data for training or pre-training, and 3D voxel grids lack temporal consistency/motion that many video segmentation works utilize. Unsupervised video object segmentation (UVOS) works mainly focus on the DAVIS dataset, which benchmarks the task of background/foreground segmentation with usually salient, moving objects. Some UVOS methods use additional labels [1, 2], while others use Mask R-CNN pre-trained on COCO [3, 4], or ir-CSN pre-trained on IG-65M and Kinetics [5]. Additionally, unsupervised action segmentation methods for video are difficult to adapt for 3D semantic segmentation because they generally rely on exploiting the temporal structure of action sequences, which does not carry over to 3D images.
>
> [1] Lu, Xiankai, et al. "See more, know more: Unsupervised video object segmentation with co-attention siamese networks." Proceedings of the IEEE/CVF Conference on Computer Vision and Pattern Recognition. 2019.
>
> [2] Wang, Wenguan, et al. "Learning unsupervised video object segmentation through visual attention." Proceedings of the IEEE/CVF Conference on Computer Vision and Pattern Recognition. 2019.
>
> [3] Luiten, Jonathon, Idil Esen Zulfikar, and Bastian Leibe. "Unovost: Unsupervised offline video object segmentation and tracking." Proceedings of the IEEE/CVF Winter Conference on Applications of Computer Vision. 2020.
>
> [4] Wang, Ye, et al. "Unsupervised video object segmentation with distractor-aware online adaptation." Journal of Visual Communication and Image Representation 74 (2021): 102953.
>
> [5] Mahadevan, Sabarinath, et al. "Making a case for 3D convolutions for object segmentation in Videos." arXiv preprint arXiv:2008.11516 (2020).
>
> [6] Siam, Mennatullah, et al. "Video object segmentation using teacher-student adaptation in a human robot interaction (hri) setting." 2019 International Conference on Robotics and Automation (ICRA). IEEE, 2019.
>
> Re: More real medical datasets
>
> We agree that more real medical datasets can be used for evaluation, and in our appendix we additionally show a qualitative example of our method’s results on a real world cryoET dataset from our own research. We use BraTS as it is the most well-known fully-annotated biomedical dataset of 3D voxel grids. Within our evaluation datasets, we include a variety of biomedical domains including 1) our biologically-inspired synthetic dataset, 2) an irregular version of the synthetic dataset with more difficulty (appendix A.5 Irregular synthetic dataset), 3) the real world BraTS dataset, as well as 4) our real world cryoET dataset (appendix A.8 Qualitative results in electron tomography).
>
> Re: Novelty
>
> We would like to clarify that we are the first to leverage hyperbolic representations for complex 3D data, as compared to simple graphs or MNIST data, as well as for the difficult tasks of segmentation as opposed to classification. These new challenges require innovative methods to learn more representation discriminability, such as our hierarchical triplet loss, multi-scale sampling scheme as well as our gyroplane convolutional layer. We demonstrate that our self-supervised hierarchical triplet loss and multi-patch sampling method on 3D voxel data shows significant improvement over our baselines, and that they are essential to capturing implicit hierarchical structure in 3D images. In addition, our gyroplane convolutional layer allows for effective mapping of 3D image data from hyperbolic to Euclidean spaces. We show the success of these methods in our ablation studies. We will update our introduction to better highlight our contributions, and thank the reviewer for the suggestion.
>
> We hope that this response has addressed the reviewer’s concerns and that they will consider increasing their rating.

---

### Official Review · Reviewer_c7iX · 2021-07-15

**Rating:** 6
**Confidence:** 3

**Summary:**

The authors present a novel representation for learning hierarchical relations within samples (3D image pixels) in an embedding space. A component that that enables this mapping is a novel self-supervised loss function. Results on real data indicate advantageous properties.

**Limitations And Societal Impact:**

Limitations: yes

Societal impact: does not apply

**Main Review:**

* Information on hierarchical representations are often ignore in multi-class classification / image segmentation tasks.
* The concept of mapping into a hyperbolic space can be considered to be innovative, as can the loss function
* Results on exemplary application data is promising.
* A discussion of how a hierarchical organization of the data can be used explicitly in this framework would be helpful.

**Time Spent Reviewing:**

1

---

> ### Author Response · Authors · 2021-08-10
> **Response to Reviewer c7iX**
>
> We thank the reviewer for pointing out that we “present a novel representation for learning hierarchical relations within samples” and that “results on real data indicate advantageous properties.” We address specific reviewer questions below.
>
> Re: Further discussion of how hierarchical nature helps
>
> Many biomedical structures have compositionality and an inherent hierarchy. For example, in cryoET images, within cells are mitochondria and aggregates, and within mitochondria there are cristae and other subcellular structures. Such types of hierarchies are present in many types of biomedical images. Capturing such structure in the data in a representation space allows for more discriminative representations that can facilitate downstream tasks such as segmentation. To efficiently and effectively capture hierarchical structure, which occurs implicitly in our 3D voxel grid input, we leverage hyperbolic representations. Our hyperbolic VAE along with hierarchical self-supervised loss implicitly learns the inherent hierarchy that is present in many biomedical structures.  We thank the reviewer for the suggestion and will clarify how hierarchical nature helps in the main text.
>
> We hope that this response has addressed the reviewer’s concerns and that they will consider increasing their rating.

---

### Official Review · Reviewer_e4nE · 2021-07-17

**Rating:** 6
**Confidence:** 3

**Summary:**

In this paper, the authors proposed to learn hyperbolic representations using 3D hyperbolic VAE with a novel gyroplane convolutional layer before 3D conv encoder. Apart from standard VAE loss, a self supervised triplet loss is also added to attract related subvolumes and repel unrelated subvolumns in the embedding space. They demonstrated their results on a biologically-inspired synthetic dataset and a brain tumor segmentation dataset, and showed that their approach achieves state-of-the-art unsupervised 3D segmentation.

**Ethical Concerns:**

No ethical concerns.

**Limitations And Societal Impact:**

No, but it seems there is no potential negative social impact

**Main Review:**

Originality: while this work can be viewed as a combination of known methods, I feel that the work contains sufficient novelty. 1) the application of the hyperbolic latent to this particular task is novel, 2) modifications to [Ganea et al, 2018] to make the gyroplane layer convolutional, a necessary step for the VAE, and 3) the self-supervised pretext task and accompanying loss. I am sold on the intuition that the hyperbolic latent has some nice properties for these types of problems.

Quality: Overall, I am impressed with the technical soundness and completeness of the work. However, I have a few questions:
- Regarding the pretext task and triplett loss: The negative patch is a spatially distant patch, but can it be content-wise similar to the positive patch as well? In Figure 1 and Table 3, we can see that both the synthetic BraTS datasets have a higher degree of symmetry than natural images. For example in Figure 2, if a positive patch is at the top left corner, is it fair to name the patch at the bottom right corner as a negative?
- The role of the upsampling and downsampling is unclear
- Why use d=2? Unless I am reading the table in the appendix wrong, you’d benefit from a higher dimensional latent.

Clarity: I think the clarity of the work is excellent overall. Great job!

Significance: I believe the task addressed here is an important one, so from this perspective I think the work has significance. The interest to the NeurIPS community would be higher if results could be shown in another domain. Also, as a sanity-check it would be good to validate your reimplementation of Naelpa et al. [2020] and Moriya et al. [2018] by reproducing their reported results in the Appendix. It would also be preferable to report multiple runs, or better yet perform statistical tests. You may want to add [1] to your comparisons.

Other comments:
- You may want to add [2] to your related work
- Interesting that gyroconv + triplet performs worse at DICE level 3… thoughts?
- The broader impact, I believe, should be have been included in the main article

[1] Uzunova, Hristina, et al. "Unsupervised pathology detection in medical images using conditional variational autoencoders." International journal of computer assisted radiology and surgery 14.3 (2019): 451-461.

[2] Baur, Christoph, et al. "Deep autoencoding models for unsupervised anomaly segmentation in brain MR images." International MICCAI Brainlesion Workshop. Springer, Cham, 2018.

**Time Spent Reviewing:**

2.5

---

> ### Author Response · Authors · 2021-08-10
> **Response to Reviewer e4nE**
>
> We thank the reviewer for pointing out that they are “impressed with the technical soundness and completeness of the work” and that our “approach achieves state-of-the-art unsupervised 3D segmentation.” We address specific reviewer questions below.
>
> Re: Pretext task and triplet loss
>
> We agree that the sampling method will not always produce reasonable pairs, but in aggregate it provides useful learning signal across different scales of sampling. Our work aims to inject as much inductive bias as we can about the relationship between spatial distance and compositionality, and we show quantitatively that our hyperbolic triplet loss is able to improve performance significantly. We will clarify in the final paper.
>
> Re: Clarifying upsampling/downsampling
>
> Using our patch sampling methodology (Section 4), the positive and negative patches can have random size. We upsample and downsample randomly sized input patches in order to use the same shared encoder in our hyperbolic VAE formulation. We will clarify in the final paper.
>
> Re: Latent dimension
>
> We use d=2 because we found that performance plateaus beyond that point. The latent dimension ablation table (A.4 Latent dimension ablation) indicates that in hyperbolic space, due to our representation space being more efficient, additional dimensions don't improve performance. In comparison, in Euclidean space, increased dimensionality does improve performance until it plateaus at d=16.
>
> Re: Additional domains
>
> We focus on biomedical datasets because they are the most common application domain that has benchmarks for 3D voxel grid data (BraTS), and significant labelled datasets are rare in other domains. Within the available datasets, we have tried to present a diverse set of domains. In our appendix, we additionally show a qualitative example of our method’s results on a real world cryoET dataset. Our evaluation datasets include 1) our biologically-inspired synthetic dataset, 3) an irregular version of the synthetic dataset with more difficulty (see appendix A.5 Irregular synthetic dataset), 3) the real world BraTS dataset, as well as 4) our real world cryoET dataset (see appendix A.8 Qualitative results in electron tomography).
>
> Re: Reproducing prior work
>
> We agree that reproducibility is highly important for the field of machine learning. As the field of 3D unsupervised segmentation is relatively unexplored, there are few papers that report unsupervised results, especially on the BraTS dataset, and the 3D unsupervised methods we found for comparison did not benchmark on the BraTS dataset. However, we have made our best efforts to ensure accuracy and fairness of the implementations we used. For Ji et al, we note that we directly use their official implementation (https://github.com/xu-ji/IIC) that is publicly available on the datasets in our paper, so it is not our own reimplementation and should be an accurate implementation of their approach. To provide further sanity check that is requested, we have begun running this implementation on the dataset in their paper, Potsdam-3, a segmentation dataset of aerial satellite images of Potsdam and is publicly available, although our other baseline methods do not report results on this dataset. The running time for training Ji et al on their dataset is projected to take several weeks, so we are unable to finish this within the scope of the author response period, but we will report results in our final appendix. In the case of Nalepa et al, we reimplemented their main contribution, the clustering layer described in the paper, in order to make a fair comparison to our encoder and decoder architecture. Moriya et al only reports results on a single dataset, which has not been released to the public, and the authors did not make their code publicly available. We use the official implementation of JULE (https://github.com/jwyang/JULE.torch), which Moriya et al is based on, to reproduce their implementation. We will also ensure that all of our code and datasets are publicly available so that our work may also serve as a useful reference for future research in this area.
>
> Re: Multiple runs
>
> We agree with the reviewer that results over multiple runs will be helpful, and provide error bars  for our method over four independent runs in the appendix (A.6 BraTS ablations, error bars, and Hausdorff distance).
>
> Re: New baseline comparison to [1]
>
> We thank the reviewer for the recommendation, and will cite the paper in our related works. The results reported in the paper are not directly comparable to the results in our paper, as they trained on the BraTS 2015 dataset and evaluation only occurred on the tumor core, not the whole tumor as in our paper. Ideally we would like to compare results on BraTS 2019, but as their codebase is not available, and their method 1) requires patches not containing pathologies for training (which isn’t completely unsupervised as in our formulation), 2) excludes images that suffer from bad resolution, and 3) multiplies all distance images with the Gaussian smoothed masks of the brain, we are unable to reproduce their results.
>
> Re: Addition to related works [2]
>
> We thank the reviewer for the recommended work, and agree that it would be useful to cite [2] in our related work as an unsupervised biomedical VAE method. We will add this citation in our final paper.
>
> Re: Combination of gyroconv + triplet
>
> We hypothesize that the positive effect of the gyroplane convolutional layer is less obvious at DICE level 3, since each object is only 2~3 pixels in width and height.
>
> Re: Broader impact in main article
>
> In the official NeurIPS 2021 FAQ, it is stated that a discussion of potential negative society impacts may appear anywhere in the paper, including in the supplemental material if appropriate. Our broader impact statement is in the supplementary in the current version of the paper. In the final paper version, we will try to add in our discussion in the main text if space allows.
>
> We hope that this response has addressed the reviewer’s concerns and that they will consider increasing their rating.

---

> > ### Comment · Reviewer_e4nE · 2021-09-01
> > **Re: author response**
> >
> > Thanks to the authors for the responses, which address most of my questions. After reading the reviews and rebuttals, I maintain my score. Although the technical novelty is limited since all main components of the proposed method already exist, I see its sufficient originality in exploring the feasibility of combining existing methods in the medical domain.

---

### Official Review · Reviewer_rQZz · 2021-07-31

**Rating:** 6
**Confidence:** 3

**Summary:**

The authors employ hyperbolic embeddings to capture implicit hierarchical relationships in biomedical imaging. Novel components of the proposed method are (1) a self-supervised triplet loss and (2) a gyroplane convolutional layer. They validated their method on (1) synthetic toy dataset and (2) BraTS'19 benchmark, where the proposed method demonstrated superior/competitve performance compared to other unsupervised/semi-supervised/supervised methods.

**Limitations And Societal Impact:**

It doesn't seem like the authors adequately addressed the limitations of the proposed method.

**Main Review:**

Overall, this paper proposes an interesting combination of hyperbolic embeddings and self-supervised learning, and applies it to biomedical imaging which is in need of advanced unsupervsed learning methods. As such, I would consider the result of this paper worth reporting despite of limited technical novelty. However, I have a major reservation on experimental validation of the proposed method.

[Originality]

Technical novelty of the proposed method is limited to (1) introducing self-supervised triplet loss (which is not novel by itself) and (2) generalizing the gyroplane layer (Ganea et al. 2018) to the gyroplane convolutional layer (novel but limited). Nevertheless, it is a novel exploration to adopt hyperbolic embeddings to capture implicit hierarchical structures in biomedical images.

[Quality]

The quality of the paper is okay, but more thorough experimental validation is required. The synthetic toy dataset is only a proof-of-concept that doesn't tell much about the validity of the proposed method. It's an interesting start point for research, but much more emphasis should be put on real world validation. One thing is that although the toy dataset is said to be inspired by cellular organization, it is not that similar to the actual cell in terms of the scale. The superficial/conceptual resemblance isn't really meaningful/helpful. There exist a lot of publicly available electron microscopy (EM) images of cells, so it would be interesting to see what the proposed method learns from EM cellular images.

Second, the validation on the BraTS'19 dataset is too limited. Reproducing other methods is non-trivial. There must be a sanity check on whether the used code (either original or reimplementation) and experimental setup reproduce the original numbers reported by the original authors. And that's why the public benchmark and its leaderboard exist. It's a *fair* battleground for competiting methods. Why didn't the authors try to use all modalities (not just FLAIR) and evaluate their method on the test set? Also, qaulitative inspection is largely missing. People would find it interesting to see a thorough qualitative inspection of the proposed method on real world datasets, especially given that the method is unsupervised and claimed to be useful for novel discovery.

[Clarity]

The paper is written clearly in general, organized well, and adequately informative.

[Significance]

It is significant to advance unsupervised learning methods for biomedical imaging. Self-supervised learning was facilitated by the application of hyperbolic embeddings to capture implicit hierarchical structures, which is another significant finding.

**Time Spent Reviewing:**

6

---

> ### Author Response · Authors · 2021-08-10
> **Response to Reviewer rQZz**
>
> We thank the reviewer for pointing out that “this paper proposes an interesting combination of hyperbolic embeddings and self-supervised learning, and applies it to biomedical imaging which is in need of advanced unsupervised learning methods” and that “it is a novel exploration to adopt hyperbolic embeddings to capture implicit hierarchical structures in biomedical images.” We address specific reviewer questions below.
>
> Re: More experimental validation on cryo-EM datasets
>
> We agree that it would be useful to have additional real world validation of our method. In our own research with ET cellular images, we ourselves are using these types of methods to study differences in anatomical structure across large patient populations to develop targeted treatment for disease based on patient anatomical characteristics. We would like to point the reviewer to our appendix, where we include examples of results from our cryoET dataset (A.8 Qualitative results in electron tomography), as well as an additional irregular synthetic dataset which better resembles cellular structure (A.5 Irregular synthetic dataset). Publicly available electron microscopy datasets, such as EMPIAR and EMDataResource, generally don’t include the 3D per-voxel annotations needed for quantitative evaluation. Our qualitative evaluations shown in the appendix are on mitochondria of neuronal cells. We hope this clarifies the reviewer’s question.
>
> Re: Reproducing other methods
>
> We agree that reproducibility is highly important for the field of machine learning. As the field of 3D unsupervised segmentation is relatively unexplored, there are few papers that report unsupervised results, especially on the BraTS dataset, and the 3D unsupervised methods we found for comparison did not benchmark on the BraTS dataset. However, we have made our best efforts to ensure accuracy and fairness of the implementations we used. For Ji et al, we note that we directly use their official implementation (https://github.com/xu-ji/IIC) that is publicly available on the datasets in our paper, so it is not our own reimplementation and should be an accurate implementation of their approach. To provide further sanity check that is requested, we have begun running this implementation on the dataset in their paper, Potsdam-3, a segmentation dataset of aerial satellite images of Potsdam that is publicly available, although our other baseline methods do not report results on this dataset. The running time for training Ji et al on their dataset is projected to take several weeks, so we are unable to finish this within the scope of the author response period, but we will report results in our final appendix. In the case of Nalepa et al, we reimplemented their main contribution, the clustering layer described in the paper, in order to make a fair comparison to our encoder and decoder architecture. Moriya et al only reports results on a single dataset, which has not been released to the public, and the authors did not make their code publicly available. We use the official implementation of JULE (https://github.com/jwyang/JULE.torch), which Moriya et al is based on, to reproduce their implementation. We will also ensure that all of our code and datasets are publicly available so that our work may also serve as a useful reference for future research in this area.
>
> Re: Only FLAIR modality
>
> While we agree that this would be helpful, using all the modalities in the BraTS dataset adds an additional dimension to our 3D input, and 4D is outside of the scope of our paper. Our work can be extended to 4D in the future.
>
> Re: Qualitative inspection
>
> Beyond our qualitative results in our main paper (4.2 Brain Tumor Segmentation challenge dataset, Table 3), we also include an additional qualitative inspection on a real world cryoET dataset in our appendix (A.8 Qualitative results in electron tomography). From inspection of results in our cryoET dataset, we see that our segmentations follow a similar trend. We will add more qualitative examples to our final paper appendix.
>
> We hope that this response has addressed the reviewer’s concerns and that they will consider increasing their rating.

---

### Decision · Program_Chairs · 2021-09-27

**Decision:**

Accept (Poster)

**Comment:**

This paper addresses the unsupervised segmentation of 3D biomedical images. The authors propose to leverage hyperbolic representations as continuous models for modeling hierarchical structures. A VAE based on Gyroplane convolutional layers at the decoder level is introduced, and trained with a self-supervised loss in addition to the ELBO data fitting based on making distance between .

The paper initially received three weak accept and one weak reject recommendation. The main concerns pointed out by reviewers were about the lack of novelty of the proposed method, and the need to compare to stronger baselines and on broader datasets. After rebuttal, all reviewers stick on their initial ratings.

The AC carefully read the submission and authors' feedback. Although the approach uses known tools (hyperbolic representation, self-supervised learning), the AC considers that its adaptation for unsupervised 3D biomedical image segmentation is meaningful. The paper is also well written and clear. The experiments are overall convincing, although the AC agrees with the reviewers' concerns related to the fact that the synthetic dataset could have been designed more thoroughly.
The AC thus recommends acceptance, but highly encourages the authors to carefully take into account and reviewers' comments and there feedback to improve the final version of the paper. Especially, the different metrics should be included in the main paper, and the author should also verify that their reimplementation of the baselines reach the performances reported in the published papers. The submission has been discussed with the senior area chair who agrees with the recommendation.